# UniMath: A Foundational and Multimodal Mathematical Reasoner

**Zhenwen Liang**[1], **Tianyu Yang**[1], **Jipeng Zhang**[2], and **Xiangliang Zhang**[✉1]

[1]University of Notre Dame, `{zliang6, tyang4, xzhang33}@nd.edu`
[2]Hong Kong University of Science and Technology, `jzhanggr@conect.ust.hk`

## Abstract

While significant progress has been made in natural language processing (NLP), existing methods exhibit limitations in effectively interpreting and processing diverse mathematical modalities. Therefore, we introduce UniMath, a versatile and unified system designed for multimodal mathematical reasoning tasks. Tackling complex problem-solving in arithmetic, geometry, and table-based math, UniMath utilizes a fine-tuned T5 model augmented with a variational autoencoder (VAE)-based image tokenizer. By jointly training and evaluating the model on three diverse datasets - SVAMP, GeoQA, and TableMWP, UniMath achieves state-of-the-art performance. The model's generalization ability is further demonstrated via fine-tuning on two additional datasets, MathQA and Geo-Proving. Through comprehensive evaluations, we show that joint training across diverse math tasks improves overall model performance and enhances its ability to generalize across different mathematical reasoning tasks. This pioneering approach provides a blueprint and inspires further efforts on unified mathematical reasoning with deep learning systems.

## 1 Introduction

Mathematical reasoning, an essential aspect of human intelligence, plays a pivotal role in our daily lives and decision-making processes (Dehaene and Sybesma, 1999). Despite the significant progress in natural language processing (NLP) and research, the development of a robust NLP system capable of handling multimodal input and accommodating diverse downstream tasks remains an under-explored challenge. Mathematical reasoning naturally involves a wide range of tasks and modalities, reflecting the complexity and adaptability of human thinking in this area.

One common task used to test mathematical reasoning skills is solving math word problems (MWP), which necessitates comprehension of textual information and execution of symbolic reasoning (Hosseini et al., 2014; Kushman et al., 2014). Another essential aspect of mathematical reasoning is geometry problem-solving, which demands the understanding of visual context and reasoning on spatial relations (Seo et al., 2015; Lu et al., 2021; Chen et al., 2022a). Furthermore, table-based math problem-solving presents a unique challenge as it requires processing heterogeneous and structured table content to extract relevant information for problem-solving (Pasupat and Liang, 2015; Lu et al., 2023a). Existing work often resorts to task-specific models, each fine-tuned for a specific modality. While these models perform efficiently in their specific domains, they struggle to generalize across different modalities, which is an essential capability for advanced AI systems. Observing this limitation, we propose a shift towards a unified model, applicable across all tasks and grounded in formal symbolic language generation. Our proposed model aims to address a wide range of tasks in the mathematical reasoning domain with a single adaptable model, improving its capacity for advanced reasoning similar to human cognition.

In the pursuit of multimodal reasoning, existing approaches can be broadly categorized into two types: 1) Lu et al. (2022) leveraging pre-trained image captioning models to convert images into textual descriptions, which are then combined with the text content, and 2) Chen et al. (2021a, 2022a) employing pre-trained image feature extractors, such as ResNet He et al. (2016), to obtain latent representations of images and concatenate them with the latent features of the text component. However, these approaches exhibit limitations when applied to mathematical reasoning tasks. Both image captioning and ResNet models are primarily designed to comprehend and describe real-world images rather than mathematical diagrams containing geometric shapes. As a result, these models

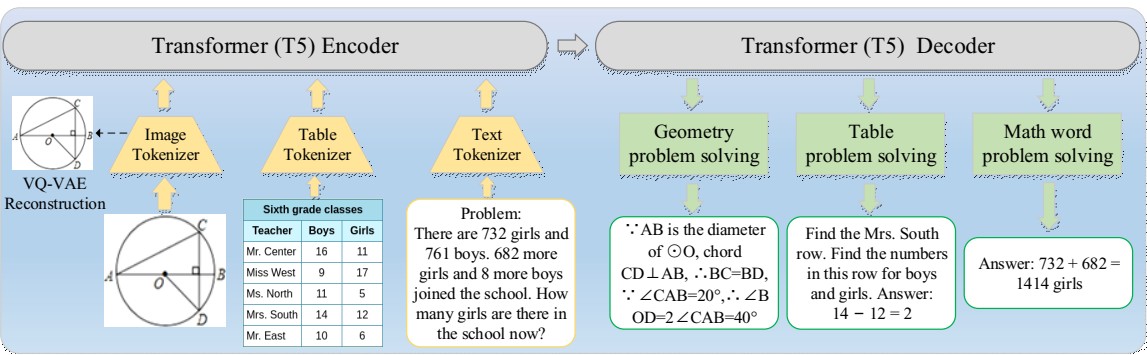

Figure 1: The overall workflow of our proposed UniMath.

may not be suitable for accurately interpreting geometry images in a mathematical context.

Another challenge in the development of a unified model for mathematical reasoning is the issue of data management, including data collection and tokenization. To achieve unified reasoning, We need to process geometric diagrams, natural languages, and symbolic representations. Handling these various data modalities, which involve tasks such as pre-processing and tokenization, emerges as a significant question of exploration. Ensuring an efficient and coherent process to unify these divergent data types is critical to the successful implementation and performance of a singular model for mathematical reasoning.

In this paper, we present UniMath, a unified mathematical reasoner capable of processing multimodal input, including math word problems (MWP), geometry problems, and table-based MWP. We achieve this by fine-tuning a T5 (Chung et al., 2022) model and augmenting its vocabulary with tokenized image representations generated by trainable VQ-VAE (Van Den Oord et al., 2017; Razavi et al., 2019). We jointly train and evaluate our model on three held-in datasets, i.e., SVAMP (Patel et al., 2021), GeoQA (Chen et al., 2021a) and TableMWP (Lu et al., 2023a). Also, we finetune our UniMath towards two held-out datasets MathQA (Amini et al., 2019) and Geo-Proving (Chen et al., 2022a) to demonstrate its potential on external tasks. Furthermore, we conduct a comprehensive evaluation of different data tokenization methods and examine the impact of chain-of-thought explanations in training such a multimodal reasoner. We believe that our work serves as a pioneering study in the development of unified mathematical reasoning systems, inspiring future researchers to devise more advanced approaches in this domain.

## 2 Our Approach

The architecture of our proposed model can be found in Fig. 1, a detailed description of our framework is located in the appendix. We build UniMath on a T5 backbone (also extendable to any encoder-decoder LLMs). For the MWP solving and table MWP solving task, we train the model with textual problem description as input and the answer as output. For tables, we follow the approach in (Lu et al., 2023a) to transform tables into texts. Also, We separate the explanation and answer of the TableMWP dataset into two targets during training controlled by different prefixes. For geometry problems, we integrate an external Vector Quantized Variational Autoencoders (VQ-VAE) as our image encoder to transform image patches to new tokens and concatenate them with the textual tokens as the input. This framework is flexible to deal with text-only input and text-image mixed input. Also, the image tokenizer employed in our approach is flexible and compatible with various backbone models, allowing for the integration of more advanced geometry feature extractors when they become available. Empirically, we use two 2-layer ResBlocks as our encoder and decoder in VQ-VAE. Our UniMath is trained by simply adding the VAE reconstruction loss and the cross-entropy loss of our pre-trained language model backbone. During testing, the decoder part of VQ-VAE is discarded.

## 3 Experiments

### 3.1 Held-in Datasets

**SVAMP** The SVAMP dataset (Patel et al., 2021) comprises 1,000 English math word problems, created by introducing challenging variations to existing problems. We adopt the original evaluation settings proposed in (Patel et al., 2021), where MAWPS (Koncel-Kedziorski et al., 2016)

|  | Held-in Tasks | | | Held-out Tasks | |
|---|---|---|---|---|---|
|  | SVAMP | GeoQA | TableMWP | MathQA | UniGeo-Proving |
| Best Fine-tuned Baseline | **47.3**[a] | 46.8[b]* | 58.5[c] | 78.6[a] | 80.6[b]* |
| Train Individually on T5-base | 29.8 | 43.7 | 62.7 | 82.3 | 82.7 |
| Train Individually on Flan-T5-base | 30.5 | 45.1 | 64.5 | 82.0 | **83.0** |
| UniMath-T5-base | 37.3 | 49.6 | 65.4 | **83.3** | 82.9 |
| UniMath-Flan-T5-base | 41.8 | **50.0** | **66.5** | 82.7 | **83.0** |

Table 1: The comparison between our unified model and baselines in terms of solution accuracy. a: Jie et al. (2022), b: (Chen et al., 2022a), c: (Lu et al., 2023a). *: Our reproduced results based on the official codes from the authors.

and ASDiv-a (Miao et al., 2020) serve as the training set and SVAMP is used as the testing set.

**GeoQA**   GeoQA (Chen et al., 2021a) contains 4,998 diverse real-world geometry problems found in Chinese middle school exams. Each problem is further annotated with specific programs that outline the problem-solving process. All problems in this dataset are of the calculation type, implying that their solutions are derived and calculated from the problem description. We utilize the English version of this dataset provided in (Chen et al., 2022a) to ensure linguistic consistency with other datasets.

**TableMWP**   The TabMWP dataset (Lu et al., 2023a) includes 38,431 tabular math word problems consisting of free-text questions and multiple-choice questions. A distinguishing feature of TabMWP is that each problem is accompanied by a tabular context, in both image format and texual format, which are essential to solve the problem.

### 3.2   Held-out Datasets

**MathQA**   MathQA (Amini et al., 2019) contains 37,200 math problems collected from GRE exams. However, many problems are either unsolvable through equations or annotated in an incorrect format. Consequently, we follow Jie et al. (2022) and select an MWP subset, comprising 16,191 problems in the training set and 1,601 for testing.

**UniGeo-Proving**   UniGeo-Reasoning (Chen et al., 2022a) features 9,543 proving problems, with each entry consisting of a colored geometry diagram, a description text, and a proof with reasons. The latter includes the reasoning skills or geometry theorems applied in each step.

### 3.3   Implementation Details

We implement our code with Pytorch framework, all the experimental results can be produced by a

single NVIDIA RTX 3090 GPU. As for hyperparameters, We utilized a batch size of 16, thereby ensuring an optimal balance between computational efficiency and model accuracy. To mitigate the effect of overfitting and help regularize the model, we implemented a dropout rate of 0.1. This allowed for a healthy proportion of neurons to be randomly ignored during training, promoting generalization and reducing potential over-dependency on certain features. In addition, we used AdamW as our optimizer with a learning rate of 0.0004 and weight decay of 0.01, aiding in optimizing our model's parameters while controlling for overfitting by preventing the weights from growing too large. To encourage the model to converge more reliably during the initial stages of training, we employed a linear warm-up with ratio of 0.1. Furthermore, to avoid exploding gradients which can result in destabilization of the training process, we employed gradient clipping with a maximum gradient norm of 5.0. This selection of hyperparameters was strategic, balancing model performance with computational resource usage.

### 3.4   Main Results

We evaluated the effectiveness of our UniMath model on two backbones, T5 and Flan-T5, and benchmarked it against the best fine-tuned baselines (we exclude those prompting-based baselines that use LLMs), as illustrated in Table 1. First, the model was jointly trained and evaluated on three held-in tasks, the results of which are shown in the first three columns. Then, to test the model's generalizability while training on out-of-distribution data, it was fine-tuned separately on two external datasets, with the results in the last two columns. To demonstrate the impact of our unified training approach, we also provide the accuracies when each dataset was separately and individually trained on the same backbone. Notably, UniMath outper-

forms the baseline in two of the three held-in tasks and in both held-out tasks. While UniMath does not beat the DeductiveReasoner (Jie et al., 2022) on the SVAMP dataset, it should be noted that the baseline model uses a specialized neural-symbolic decoder, which typically outperforms a general-purpose decoder like T5, albeit lacking generalizability to non-MWPs. The main takeaways are:

1. We successfully obtain an effective unified mathematical reasoner with very competitive accuracy against state-of-the-art baselines.

2. The mathematical reasoning ability derived from held-in tasks is able to generalize and help improve the fine-tuning on held-out tasks.

### 3.5 Analysis on Symbol Pre-processing

Mathematical problem-solving often involves various symbolic representations such as $+, -, \triangle, \perp, \cong, \simeq$. Pre-processing these special tokens has been a topic of interest in recent research (Wang et al., 2017; Zhang et al., 2020b; Jin et al., 2021; Chen et al., 2022a). We examined the pre-processing techniques of two categories of symbols: arithmetic operations (i.e., $+, -$) and geometric relations (e.g., $\perp$ and $\cong$). In Table 2, we show a comprehensive list of symbol-to-name transformations. *We considered four different settings: 1)*

| Category | Symbol | Representation |
|---|---|---|
| Arithmetic | + | cal_add |
| Arithmetic | - | cal_minus |
| Arithmetic | * | cal_multiply |
| Arithmetic | / | cal_divide |
| Geometric | $\neq$ | not_equal |
| Geometric | $\approx$ | approximate |
| Geometric | $\triangle$ | triangle |
| Geometric | $\angle$ | angle |
| Geometric | $\parallel$ | parallel |
| Geometric | $\odot$ | circle |
| Geometric | $\perp$ | perpendicular |
| Geometric | $\cong$ | congruent |
| Geometric | $\square$ | parallelogram |
| Geometric | $\sim$ | similar |
| Geometric | $\frown$ | arc |

Table 2: The symbol-to-name transformations used in our paper. We transform all geometric relations during data pre-processing to help the language model understand them.

*no transformation, 2) transform arithmetic operators only, 3) transform geometric relations only,*

*4) transform both.* As evidenced by the results in Figure 2(a), arithmetic operators perform optimally when preserved in their original form, while geometric symbols are better when transcribed into their natural language counterparts such as *perpendicular_to* and *congruent_to* like function names. This discrepancy may stem from the higher frequency of arithmetic operators in the pre-training corpus, thereby enabling models like T5 to have a more nuanced understanding of them compared to the less common geometric symbols. In our UniMath tokenizer, we only transform geometric relations according to the results of our analysis in this section.

### 3.6 Analysis on Image Tokenizer

Geometry problem-solving necessitates models capable of deciphering geometric diagrams. Existing techniques, such as those referenced in (Chen et al., 2021a, 2022a), employ frozen ResNet-101 to derive embeddings by treating diagrams as images. However, given the significant differences between geometric diagrams and real-world images, the suitability of ResNet is questionable. In this study, documented in Figure 2(b), we carry out a thorough examination of various image tokenizer designs. *We considered five different settings: 1) Frozen ResNet, 2) Trainable ResNet, 3) Two trainable Resblocks, 4) Trainable ResNet + VQ-VAE, and 5) Two trainable Resblocks + VQ-VAE.* Notably, while the ResNet was pre-trained on ImageNet, the Resblocks were randomly initialized. Additionally, it's worth mentioning that two Resblocks have significantly fewer parameters than ResNet. Our findings reveal that ResNet coupled with VQ-VAE yields the most impressive results, though two ResBlocks in conjunction with VQ-VAE can achieve comparable performance with much fewer parameters. This result underscores the beneficial impact of VQ-VAE reconstruction loss in enhancing image tokenizer efficacy. Also, we can conclude that the pre-trained ResNet does not contribute as effectively to geometric diagram comprehension.

### 3.7 Analysis on Chain-of-Thought Explanations

One of our held-in tasks, namely TableMWP, is originally annotated with Chain-of-Thought (CoT) (Wei et al., 2022) explanations (i.e., step-by-step descriptions of solutions). *We evaluated performance across three different settings: 1) Omitting*

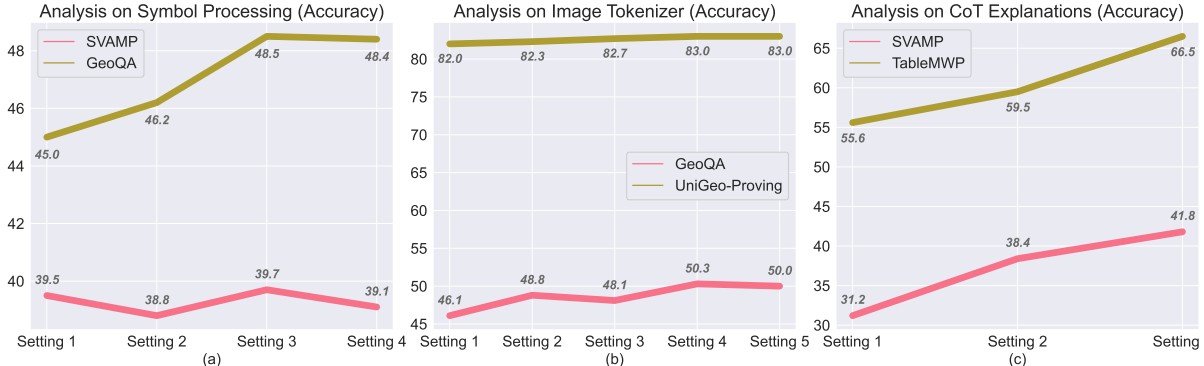

Figure 2: Detailed analysis on pre-processing, tokenizers, and chain-of-thought explanations. The descriptions of different settings are in *italics* in Section 3.5, 3.6 and 3.7.

*CoT and focusing solely on the final answer; 2) Combining CoT with the final answer to form a joint target; and 3) Generating CoT and the final answer with different prefixes, essentially splitting them into two tasks.*

Our experimental findings, detailed in Figure 2(c), indicate that incorporating the CoT method can considerably improve the problem-solving accuracy, not only on the TableMWP dataset but also on SVAMP, a dataset not originally annotated with CoT. This suggests a promising capacity for CoT to be generalized to external datasets. Additionally, our results show that multi-task learning tends to be the most effective approach compared to the other two strategies we examined.

## 4  Related Work

**Math Word Problem Solving**: Math word problem solving is a prevalent evaluation method for NLP models (Amini et al., 2019; Patel et al., 2021; Cobbe et al., 2021). Early solutions relied on statistical and rule-based parsing (Hosseini et al., 2014; Koncel-Kedziorski et al., 2015), and have evolved into Seq2Seq-based neural networks (Xie and Sun, 2019; Zhang et al., 2020a; Liang et al., 2022a,b) and LLM-enhanced solvers (Liang et al., 2023a,b).
**Geometry Problem Solving**: Geometry problem solving often necessitates the understanding of visual diagrams and textual descriptions. State-of-the-art models and datasets, namely Geometry3K (Lu et al., 2021), GeoQA (Chen et al., 2021a), GeoQA+ (Cao and Xiao, 2022), and UniGeo (Chen et al., 2022a), are aiming to enhance both performance and explainability.
**Structural Problem Solving**: Structural math problems, especially those entailing tables, require an intricate blend of interpretation and reasoning.

For example, TAPAS (Herzig et al., 2020) introduced a new way of parsing tabular data using pre-training on a large corpus of tables. Lu et al. (2023a) introduced a new dataset and employed reinforcement learning to select in-context examples.

For a more detailed summary of mathematical problem solving, please refer to the survey (Lu et al., 2023b).
**Multimodal Foundation Models**:  Pioneering works such as BEiT (Bao et al., 2022) and Uni-perceiver (Zhu et al., 2022) have been investigating pre-training strategies for handling multimodal data. Further, all-in-one Transformer (Wang et al., 2023) and OFA (Wang et al., 2022) have shown promise in providing a unified approach to handling diverse modalities. ML-MFSL (Najdenkoska et al., 2023) introduced meta-learning that enables flexible adaptation across vision and language tasks.

## 5  Conclusion

In this work, we introduce UniMath, a unified mathematical reasoner, aiming to address diverse and multi-modal mathematical tasks.  Our extensive experiments demonstrate its superior performance and generalizability across various mathematical modalities, outperforming many task-specific baselines. Moreover, we share key insights on data pre-processing and tokenization techniques to achieve our unification goal. We further discuss the impact of chain-of-thought explanations in the training process of such a unified reasoner. We hope that this work will encourage further exploration in NLP, and contribute to the broader goal of achieving human-like mathematical reasoning in AI systems.

Codes and data are available at https://github.com/Zhenwen-NLP/UniMath.

## Limitations

Our UniMath model, while effective, does have some limitations that need to be addressed. First, we currently use convolutional neural networks, specifically ResNet or Resblocks, to process geometry diagrams. Although this is a common approach in previous studies, it might not be the best solution for every situation. In other words, there might be a more effective way to handle geometry diagrams. In our future work, we plan to incorporate a more specialized tool, a geometric parser, into our model to improve how it handles these tasks. Secondly, UniMath is built around an encoder-decoder structure, similar to the T5 model. The adaptability of our approach on popular decoder-only models, like BLOOM (Scao et al., 2022) and LLaMA (Touvron et al., 2023) models, is still unknown. In future work, we intend to explore how we can add more tokenizers to these decoder-only models.

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

# A  Appendix

## A.1  Our Approach

Our proposed method builds upon the T5 model series (Raffel et al., 2020; Chung et al., 2022), an encoder-decoder transformer architecture that has shown promising results in various natural language processing tasks. The encoder receives inputs and the decoder produces results. To enable the encoder being able to understand geometry diagrams, we integrate an external Vector Quantized Variational Autoencoder (VQ-VAE) (Van Den Oord et al., 2017; Razavi et al., 2019) as our image encoder. This allows us to transform image patches into tokens, which can then be concatenated with the textual tokens to form the input for UniMath. Because VQ-VAE uses a discrete latent space to encode the image patches, which can be efficiently processed by the T5 model.

For the tabular input, we adapt the approach of (Lu et al., 2023a) to transform tables into textual representations. Tables can be complex to parse and understand directly due to their multidimensional nature and various possible formats. By converting the tables into text, we can leverage the existing language understanding capabilities of the T5 model to solve these problems. Specifically, we convert the raw table text into a flattened token sequence, with each row separated by a newline character '\n' and each column separated by '|'. Once the table is transformed into this textual format, it can be input into UniMath in the same manner as regular problem descriptions. Also, in the TableMWP dataset, every problem is annotated with a step-by-step chain-of-thought (CoT) explanation and the final answer. We apply a multi-task learning way to separate the CoT and the answer, which is accomplished by using different prefixes to control the output of the model, enabling the model to generate both the problem solution and the explanation for the solution. An analysis of this can be found in Section 3.7.

### A.1.1  VQ-VAE Implementation

In our implementation of VQ-VAE, we use a series of two 2-layer ResBlocks as our encoder and decoder, and each ResBlock is:

$$y = \text{ReLU}(\text{Conv2}(\text{ReLU}(\text{Conv1}(x))) + x)$$

where $x$ and $y$ are the input and output, respectively.

The loss function we use for training the VQ-VAE follows the one proposed in (Van Den Oord et al., 2017). This loss function can be formulated as the addition of the following three terms:

$$L_{\text{reconstruction}} = ||x - \text{Decoder}(z_q)||_2^2$$
$$L_{\text{quantization}} = ||sg[z_e - z_q]||_2^2$$
$$L_{\text{commitment}} = \beta * ||z_e - sg[z_q]||_2^2$$

where $x$ represents the input to the model, which is the geometry diagram in our model. $z_e$ is the output from the encoder, which is a continuous latent representation. $z_q$ is the selected quantized vector that is closest to $z_e$ in the codebook of vectors. The symbol $sg$ denotes the stop-gradient operation. This operation is used in the computation of the quantization loss and commitment loss. When the stop-gradient operation is applied to a variable, during backpropagation, no gradient will be backpropagated through this variable. This means that the stop-gradient operation prevents its input from being updated by gradient descent during training. The reconstruction loss trains the image encoder and decoder to reconstruct the input image. The quantization loss encourages the quantized embedding to move towards the latent feature generated by the encoder. The commitment loss encourages the output of the encoder to stay close to the chosen embedding vector, preventing the encoder's output from fluctuating too frequently between different code vectors.

### A.1.2  Training and Testing

The training of UniMath is done by combining the VAE reconstruction loss with the cross-entropy loss of our T5 backbone. This results in a hybrid loss function that optimizes both the language understanding and image encoding capabilities of UniMath. During the testing phase, we only retain the encoder part of the VQ-VAE. The decoder part, which is used during training to reconstruct the original image, is discarded.

To sum up, our UniMath model is a comprehensive approach that handles various types of math problems, including text-based problems, table-based problems, and geometry problems, by combining the power of a transformer-based language model with a VQ-VAE-based image encoder. This combination allows UniMath to effectively interpret and solve a wide range of math problems.