# OpenReview forum: "UniMath: A Foundational and Multimodal Mathematical Reasoner"
_EMNLP/2023/Conference — EMNLP 2023 Main_

### Official Review · Reviewer_xULr · 2023-08-03

**Soundness:** 3

**Excitement:**

4: Strong: This paper deepens the understanding of some phenomenon or lowers the barriers to an existing research direction.

**Paper Topic And Main Contributions:**

This paper constructs a math-problem solving model that can handle problems with both tables and geometric diagrams. They fine-tune T5 on the training set of several math datasets and use a VQ-VAE to embed in images as tokens for geometric problems.

**Questions For The Authors:**

Did the authors try using a large VQ-VAE pre-trained on natural images from prior work? Was is necessary to train a new (and much smaller) one?

Why is the GeoQA results so low?

**Reasons To Accept:**

- Reasonable results on these benchmarks despite using a relatively small LM (T5-Base).
- Shows the benefit of multi-tasking these tasks despite them being non-trivially different, and the ability to transfer for TableMWP CoT to other tasks.
- I found use of the VQ-VAE interesting, while a lot of works uses various image backbones to embed images for LLMs using a VQ-VAE is less well studied.

**Reasons To Reject:**

- I am a bit confused by their comparisons. Chen et al report 60% on GeoQA, why is number here so much lower? Because the authors were not able to reproduce their result? Le et al. report 68% on TableMWP, I assume that number has been not shown because its not considered a "fine-tuned baseline" but I think for honesty's sake it should be in the table.
- The abstract and intro state "UniMath achieves state-of-the-art performance" but I as far as I can tell existing work has reported better scores on SVAMP, GeoQA, and TableMWP so this claim seems to be simply false.

I feel like this paper could be accepted since I thought some of the scientific finding were interesting and being behind other works that use much larger LLMs is not unexpected. However the abstract and intro over-promises and the main results table is questionable in how to presents prior work to the point I would call this paper misleading.

The rebuttal has significantly mitigated this concern for me, so I have raised my score.

**Reproducibility:**

4: Could mostly reproduce the results, but there may be some variation because of sample variance or minor variations in their interpretation of the protocol or method.

**Reviewer Confidence:**

3: Pretty sure, but there's a chance I missed something. Although I have a good feel for this area in general, I did not carefully check the paper's details, e.g., the math, experimental design, or novelty.

---

> ### Author Rebuttal · Authors · 2023-08-23
>
> Thank you for taking the time to review our paper and providing such detailed feedback. We are also very grateful for confirming that our paper has some interesting findings. We sincerely hope the reviewer can raise the assessment score of our paper if the following responses have successfully addressed the concerns.
>
> **Concerning GeoQA Comparisons:**
> We would like to clarify that our results we presented are based on the Top-1 accuracy, while baselines [1] and [2] reported their best results using the Top-10 accuracy, where [1] actually used a beam size of 10 as a representation of top-10 accuracy. We chose the Top-1 accuracy as we believe this metric better reflects the precision required in mathematical problem solving. Nevertheless, this feedback is very helpful and has prompted us to recognize the need for clearer clarification. In our revised version, we will ensure to add both the clarification regarding our choice of accuracy metric and the comparison with baselines while using top-10 accuracy.
>
> [1] GeoQA: A Geometric Question Answering Benchmark Towards Multimodal Numerical Reasoning. Chen et, al.
>
> [2] UniGeo: Unifying Geometry Logical Reasoning via Reformulating Mathematical Expression. Chen et, al.
>
> **Concerning TableMWP Results:**
> We totally agree with the reviewer. We will revise our expressions to avoid any over-claims and include the performance of LLMs and in-context learning methods for the readers' convenience of reference.
>
> **Regarding the use of VQ-VAE pre-trained on natural images:**
>
> If we understand correctly, we have conducted this exploration in Section 3.3:
>
> `Our findings reveal that ResNet coupled with VQ-VAE yields the most impressive results, though two ResBlocks in conjunction with VQ-VAE can achieve comparable performance with much fewer parameters. This result underscores the beneficial impact of VQ-VAE reconstruction loss in enhancing image tokenizer efficacy. Also, we can conclude that the pre-trained ResNet does not contribute as effectively to geometric diagram comprehension.`
>
> We will further make our conclusions clearer and clarify our explanations in the updated version.

---

### Official Review · Reviewer_FwBq · 2023-08-05

**Typos Grammar Style And Presentation Improvements:** Not found.
**Soundness:** 4

**Excitement:**

4: Strong: This paper deepens the understanding of some phenomenon or lowers the barriers to an existing research direction.

**Missing References:**

The below work is also about the unification of math reasoning tasks and should be cited:
1. Matcha: enhancing visual language pretraining with math reasoning and chart derendering

**Paper Topic And Main Contributions:**

This paper proposes a framework which unifies various types (different in domains and input modality) of math-related generation tasks into T5-based transformer encoder-decoder framework. With the help of unification, the framework called UniMath leverages pretrained T5/Flan-T5 and jointly trained with geometry problem solving, table problem solving and math word problem solving tasks. The experimental results show the effectiveness of UniMath framework, which surpasses baselines individually trained on each downstream task, and the model is capable to be further transferred to held-out math-related tasks. Comprehensive analysis on image encoding, CoT setting, math symbol processing are also provided. In general, it is a great work towards a unified solution to math-related tasks.

**Questions For The Authors:**

Does the jointly trained UniMath model have some experimental evidence on zero-shot (or few-shot) math-related task transferring?

**Reasons To Accept:**

1. The research question is meaningful: it is important and needed for a math model to deal with different formats and modalities for the existing math-related tasks.
2. The approach of UniMath is straightforward: it's natural to format these tasks into seq-to-seq paradigm and use appropriate encoder to encode each modality (leveraging pretrained encoders).
3. The experimental result is encouraging: the joint training of multiple downstream tasks after unification is helpful for not only held-in but held-out math tasks.
4. Comprehensive analysis over various implementation details are provided.

**Reasons To Reject:**

1. More case studies are needed to demonstrate the effectiveness of UniMath, facilitating better math reasoning by jointly training.
2. For the input image modality, more details about image resolution setting and its effect on model performance should be provided.

**Reproducibility:**

5: Could easily reproduce the results.

**Reviewer Confidence:**

4: Quite sure. I tried to check the important points carefully. It's unlikely, though conceivable, that I missed something that should affect my ratings.

---

> ### Author Rebuttal · Authors · 2023-08-24
>
> Thank you for your valuable feedback.
>
> **Case Study**
>
> We will include a comprehensive case study section in the revised version of the paper to more effectively demonstrate the capabilities of UniMath in facilitating better mathematical reasoning through joint training.
>
> **Image Tokenizer Detail**
>
> We follow the settings from [1]. Specifically, we fill geometric diagrams with a white background to equalize length and width, then resize them to 224×224 pixels. These images are further segmented into 49 patches, each of 32×32 pixels. More details on this will be added to the revised paper for clarity.
>
> [1] UniGeo: Unifying Geometry Logical Reasoning via Reformulating Mathematical Expression. Chen et, al.
>
> **zero-shot or few-shot transferring**
>
> Actually, we have conducted evaluations of UniMath trained on SVAMP, GeoQA, and TableMWP across MathQA and Geo-Proving datasets. It appears that our fine-tuned UniMath model is already overfitted on the training datasets, making it challenging to generalize to other tasks in zero/few-shot settings. Nonetheless, our results indicate that UniMath can be further trained on held-out datasets to unlock its potential for out-of-distribution mathematical reasoning.
>
> We've also incorporated the suggested reference into our paper. Thank you for the recommendation!

---

### Official Review · Reviewer_U2A1 · 2023-08-06

**Soundness:** 3

**Excitement:**

3: Ambivalent: It has merits (e.g., it reports state-of-the-art results, the idea is nice), but there are key weaknesses (e.g., it describes incremental work), and it can significantly benefit from another round of revision. However, I won't object to accepting it if my co-reviewers champion it.

**Paper Topic And Main Contributions:**

This paper proposes UniMath, a multimodal mathematical problem-solving model that can solve arithmetic, geometry, and table-based math problems.

**Questions For The Authors:**

If a math problem requires more than one modality for its solution, can it be successfully solved by UniMath?
Can UniMath be deployed on decoder-only LLMs?

**Reasons To Accept:**

Solving math problems is an important task and I think that a multimodal math reasoning model can be useful as certain complex math problems demand the utilization of multiple modalities for their solution

**Reasons To Reject:**

The method seems to be a basic combination of three individual modalities (geometry, table, text).
The technical details are unclear as the authors did not provide comprehensive information regarding the model architecture, parameter size, and the training process.

**Reproducibility:**

3: Could reproduce the results with some difficulty. The settings of parameters are underspecified or subjectively determined; the training/evaluation data are not widely available.

**Reviewer Confidence:**

3: Pretty sure, but there's a chance I missed something. Although I have a good feel for this area in general, I did not carefully check the paper's details, e.g., the math, experimental design, or novelty.

---

> ### Author Rebuttal · Authors · 2023-08-23
>
> Thank you for taking the time to review our paper and for your insightful comments. We genuinely appreciate your feedback. We hope our responses can address your concerns so that you may consider increasing the scores of the review.
>
> **Reply to reason to reject**
>
> We totally understand the concerns the reviewer has raised. In fact, we have provided an exhaustive description of our training process, configurations, and technical details in the appendix of the paper. Given the constraints of space, we had to put them in the appendix in our submission. However, we would be more than happy to relocate them to the main body of our paper in our camera-ready version.
>
> Moreover, to ensure reproducibility, we have submitted our code for reference during the review process. We would like to assure the reviewers and the community that we are committed to releasing our data and codes upon the paper's acceptance as promised in our submission.
>
> **Response to the Question:**
>
> Thanks for the insightful questions. Our response:
>
> 1. Our UniMath can only output text modality. However, it can be combined with additional generative models to output other modalities, in a post-processing way.
>
> 2. It can theoretically adapt to the decoder-only LLMs by adding additional tokenizers, but it falls beyond the scope of the current paper.
>
> While we acknowledge the importance of these questions, they fall beyond the scope of our current paper. We will mention them in the limitation section of our paper and explore them in our future work.

---

### Meta-Review · Area_Chair_Rdjp · 2023-09-15

**Recommendation:** 5

**Metareview:**

This work addresses the task of multi-modal math problem solving. They augment a T5 model with a VQ-VAE image tokenizer and train the model on 3 math datasets which also include images and tables. Their model shows good performance on these datasets and also performs well when fine-tuned and evaluated on 2 other datasets.

The reviewers mostly agree this work is exciting. The initial reviews noted that some of the claims made are not accurate - particularly the depiction of existing works and reporting of the results from prior works. The authors have clarified this in the subsequent rebuttal and have promised to tone down the claims of state of the art. The authors should also report results from other works including some references that should have been included in their tables and are missing. One reviewer has given a lower score stating that the technical details were lacking and the work may not be reproducible but the authors promise to make their code available and have included information in the supplement.

Overall the contribution is good for a short paper. The authors should incorporate the feedback from the reviewers' comments to improve the paper and in particular should tone down the claims and represent results from existing works clearly.

---

### Decision · Program_Chairs · 2023-10-07

**Decision:**

Accept-Main

**Comment:**

This work addresses the task of multi-modal math problem solving. They augment a T5 model with a VQ-VAE image tokenizer and train the model on 3 math datasets which also include images and tables. Their model shows good performance on these datasets and also performs well when fine-tuned and evaluated on 2 other datasets.

The reviewers mostly agree this work is exciting. The initial reviews noted that some of the claims made are not accurate - particularly the depiction of existing works and reporting of the results from prior works. The authors have clarified this in the subsequent rebuttal and have promised to tone down the claims of state of the art. The authors should also report results from other works including some references that should have been included in their tables and are missing. One reviewer has given a lower score stating that the technical details were lacking and the work may not be reproducible but the authors promise to make their code available and have included information in the supplement.

Overall the contribution is good for a short paper. The authors should incorporate the feedback from the reviewers' comments to improve the paper and in particular should tone down the claims and represent results from existing works clearly.